# Structure-Activity Relationship Study of Majusculamide D: Overcoming Metabolic Instability and Severe Toxicity with a Fluoro Analogue

**DOI:** 10.3390/md22120537

**Published:** 2024-11-29

**Authors:** Xiuhe Zhao, Xiaonan Xi, Mingxiao Zhang, Mengxue Lv, Xiang Zhang, Yaxin Lu, Liang Wang, Yue Chen

**Affiliations:** 1The State Key Laboratory of Medicinal Chemical Biology, College of Pharmacy, Nankai University, Tianjin 300350, China; 15522215338@163.com (X.Z.); x18716062989@163.com (M.Z.); 15615637301@163.com (M.L.); 2College of Chemistry, Nankai University, Tianjin 300071, China; 2120231187@mail.nankai.edu.cn (X.Z.); yaxinlu@nankai.edu.cn (Y.L.); 3Haihe Laboratory of Sustainable Chemical Transformations, Tianjin 300192, China

**Keywords:** structure-activity relationship, natural products, anticancer, pancreatic cancer, metastasis

## Abstract

Majusculamide D, isolated from the marine cyanobacterium *Moorea producens*, is an anticancer lipopentapeptide consisting of fatty acid, tripeptide, and pyrrolyl proline moieties. In this work, by utilizing a convergent synthetic approach, late-stage modification, and bioisostere strategy, 26 majusculamide D analogues were synthesized, and two (**1i** and **1j**) demonstrated IC_50_ values < 1 nM against PANC-1 cancer cells. The results summarized a preliminary structure-activity relationship mainly at the C23, C4, C34, and C10 sites. A series of in vitro assays, including wound healing, transwell, clone formation, EdU, and western blot, confirmed that majusculamide D inhibited the migration, invasion, and proliferation of pancreatic cancer cells. The optimized fluorinated analogue **1n** demonstrated a notable enhancement in stability during the mouse plasma assay (>50% left after 24 h), exhibited tumor-suppressive effects (51.5% at a dosage of 5 mg/kg), and successfully mitigated the severe toxicity (no mouse dead) observed in the group treated with majusculamide D (3 mice dead) in a xenografted mouse model.

## 1. Introduction

Pancreatic cancer, with a 5-year survival rate of merely 12.8% [1], remains a lethal disease [2,3] without promisingly increased life expectancy. Moreover, it is likely to become the second leading cause of cancer-related mortality in America by 2030 [2]. Pancreatic cancer patients who undergo surgical resection have a chance for a cure, and subsequent adjuvant chemotherapy has improved their prognosis [4]. Nonetheless, approximately 80–85% of pancreatic cancer patients are unresectable or show cancer metastasis [4]. Chemotherapy remains the primary treatment modality to palliate cancer-related symptoms and extend life [4]. Clinically, novel drugs for treating pancreatic cancer are in urgent demand.

Developing natural product-derived bioactive entities, including small molecules and conjugates (e.g., antibody-drug conjugates), is vital for cancer therapy [5,6,7]. Furthermore, natural product-derived probes serve as effective tools for mechanistic research, enabling the investigation of complex cancer pathology [8]. Marine cyanobacteria provide kinds of small molecules with promising anticancer properties [9,10]. Microcolins and majusculamide D, isolated from the marine cyanobacterium *Moorea producens* (formerly named *Lyngbya majuscula*), share analogous structures and demonstrated bioactivities, especially for anticancer activities [11,12,13,14,15,16]. In 2019, Gerwick et al. discovered that majusculamide D exhibited remarkable inhibitory activity and up to 4 × 10^3^ selectivity against pancreatic cancer cells in vitro [17]. Meanwhile, they achieved a breakthrough by completing the first synthetic route to majusculamide D, which further facilitated the determination of its absolute configuration [17]. Wang et al. uncovered that microcolin H induced autophagic cell death and exhibited anti-proliferating activity by targeting PITPα/β [18]. Recently, the synthesis of majusculamide D, along with its eight analogues modified at the fatty acid moiety, the C10-OH group, and the C2–C3 unsaturated double bond, was completed [19]. The in vitro bioactivity data against PANC-1 cells revealed a preliminary structure-activity relationship (SAR) that C10-OH and C2–C3 unsaturated double bonds were crucial in maintaining the inhibitory activity while the fatty acid moiety was tolerable [19]. Moreover, majusculamide D was found to show inhibitory activities against several cancer cell lines [19]. Given the remarkable selectivity and potency of majusculamide D, further pharmacological and medicinal studies may offer new insights into its associated mechanistic signaling pathways and lead to the development of majusculamide D-derived candidates with enhanced activity. Herein, the syntheses of 26 majusculamide D analogues, primarily modified at the tripeptide moiety and pyrrolyl proline moiety, along with their in vitro and in vivo biological evaluations, are reported.

Retrosynthetically, the same convergent strategy previously employed by Gerwick was adopted [17]. The target lipo pentapeptides (**1**) were deconstructed into the fatty acid, tripeptide, and pyrrolyl proline moieties, which were combined through well-established amide condensation (Figure 1). To enhance the metabolic stability, it was anticipated that the C23-OAc (R^1^) of the tripeptide moiety would be replaced using the bioisostere strategy (e.g., NAc) [20]. The fluorine atom was considered to be introduced as it makes an impact on the potency, conformation, metabolism, and membrane permeability [21]. The *para*-substituted OMe (R^3^) group in the tyrosine of the tripeptide was envisioned to be changed to F, H, OH, or Ph to evaluate its impact on the bioactivities. Regarding the pyrrolyl proline moiety, the configuration of C4 was intended to be switched, or the methyl group at C4 was to be directly trimmed, to investigate whether the naturally occurring C4-*R*-Me is necessary. Given the poor water solubility of majusculamide D, we also aimed to improve its water solubility. The comprehensive SAR could guide us in further structural simplification and optimization [22].

## 2. Results and Discussion

### 2.1. Syntheses and Inhibitory Activities of Majusculamide D Analogues

Syntheses toward the target compounds are depicted in Figure 2 and Figure 3. Moreover, the reaction conditions in these schemes closely parallel those previously reported by us. Compound **6c** was prepared employing the silver-mediated trifluoromethylation of alcohol developed by Qing et al. [23]. Meanwhile, the *N*-substituted R^1^ at C23 in **6e** and **6f** was introduced via the Mitsunobu reaction using *L*-*allo*-threonine as the starting material. Following deprotection of the Boc group in compounds **6a**–**6f** with 4 N HCl (Figure 2A), the resulting amines were coupled with compounds **7a**–**7e** employing a well-established coupling condition (HATU and DIPEA) to deliver **8a**–**8j**. Subsequent debenzylation of compounds **8a**–**8j** under the catalytic hydrogenation (Pd/C and H_2_) condition provided 10 tripeptide analogues **3a**–**3j**. Next, the method developed by Gerwick [17] was employed to rapidly construct the pyrrolyl proline analogues **4a**–**4d** through a four-step sequence, which included activation of the carboxylic acid, imide formation, α-selenylation, and oxidation-triggered elimination (Figure 2B).

With building blocks in hand, we moved on to their coupling. The connection of the tripeptide and pyrrolyl proline moieties successfully delivered 13 advanced precursors, which were subsequently subjected to the final coupling with the fatty acid moiety **2a** to give compounds **1a**–**1m** (Figure 3A). Direct deprotection of the TBDPS group of compound **1b** furnished **14a** (Figure 3B) with an exposed hydroxy group at C23. Moreover, treating compound **1f** under the mild condition (Et_2_NH) removed the Fmoc group, and the resulting amine was reacted with methyl isothiocyanate to produce a thiourea compound **14b**. The removal of the allyl group from compound **1g** resulted in the formation of phenolic compound **14c.**

For preliminary evaluation of anti-pancreatic cancer effects, 12 majusculamide D analogues (see Table 1) were tested against PANC-1 cancer cells. Meanwhile, it was noteworthy that the inhibitory activities of **14a** with C23-OH dropped dramatically (IC_50_ = 1162 nM), while the OCF_3_ (**1c**), NAc (**1e**), and thiourea (**14b**) analogues, as well as the C23-demethyl analogue (**1d**), exhibited promising IC_50_ values ranging from 1.06−2.94 nM. These results indicated that the replacement of OAc with other groups at C23 was tolerable, while the exposed C23-OH deteriorated the inhibitory activity. When the methyl group of the *O*-Me tyrosine segment was removed (**14c**), the IC_50_ value of **14c** was 5.31 nM. Replacing the *O*-Me tyrosine with phenylalanine (**1h**), the IC_50_ value significantly decreased to 72.02 nM. Given that the O-Me group in anisole is a potentially metabolically unstable site, the OMe group with fluorine (**1i**) or phenyl (**1j**) were replaced, whose IC_50_ values maintained a nanomolar level (0.41 nM for **1i** and 0.70 nM for **1j**). As for the pyrrolyl proline moiety, compound **1k**, with the C10-OH on proline replaced by a fluorine atom, exhibited an IC_50_ value of 2.55 nM. Interestingly, the activity of C4 demethyl compound **1l** dropped significantly (134.2 nM), while the C4-*S* configured compound **1m** exhibited a comparable IC_50_ value (4.68 nM), demonstrating that the methyl group at C4 could not be removed directly.

### 2.2. Syntheses and Inhibitory Activities of Majusculamide D Analogues Conjugated with One Extra Amino Acid

With ample amounts of the key intermediate **1f** available, attention was also directed towards late-stage modifications to enhance the water solubility of majusculamide D by attaching amino acids [24]. The intermediate prepared from the deprotection of compound **1f** was reacted with various protected amino acids in the presence of HATU and DIPEA to give **15a**–**15m**, which were ultimately converted into compounds **16a**–**16m** through treatment with TFA. (Figure 4).

Compared to majusculamide D (**1a**), compounds **16a**–**16m** exhibited improved water solubility (Table 2). Among them, compound **16d** (solubility: 0.81 mg/mL) exhibited the highest water solubility. However, the IC_50_ value of compounds **16a**–**16m** decreased to deca−multihecto nanomolar levels. It is speculated that the addition of an extra amino acid may alter the spatial conformation, potentially affecting the ability to bind to the target or reducing membrane permeability, thereby leading to a decrease in activity.

### 2.3. Synthesis, Inhibitory Activity, Metabolic Stability Assay, and In Vivo Antitumor Activity of Majusculamide D Analogue ***1n***

According to the results of SAR and previous research, a target molecule (**1n**) was further designed in which the C23-OAc and C34-OMe groups were replaced with OCF3 and F groups, respectively, and one methyl group at C40 in the fatty acid chain was retained. Compound **1n** was successfully synthesized based on the previous procedure (Figure 5). The biological results demonstrated that the IC_50_ value of compound **1n** against PANC-1 cells was 3.83 nM.

In the meantime, a series of in vitro assays, including wound healing, transwell, clone formation, EdU, and western blot, confirmed that majusculamide D inhibited the migration, invasion, and proliferation of pancreatic cancer cells and may function by suppressing the Wnt signaling pathway and the EMT process [25,26,27,28,29,30,31] (for experimental details, see the Appendix A, Biological experiment part, and Appendix A). To determine whether the replacement of C23-OAc enhances metabolic stability, metabolic stability assays were conducted using **1a, 1c**, and **1n** in mouse plasma (Figure 1A). The results indicated that natural product **1a** vanished rapidly, with 54.7% left after 2 h. The metabolic instability may hinder its further in vivo medicinal applications. In sharp contrast, compounds **1c** and **1n** demonstrated significantly superior stability compared to **1a**, particularly **1n**, which retained more than half of its presence in the mouse plasma assay even after 24 h.

Furthermore, a xenografted murine subcutaneous tumor model was employed to evaluate the effects of majusculamide D and the optimized analogues **1c** and **1n** (Figure 1B). PANC1-bearing BALB/c nude mice were separated randomly, with a total of six mice in each group. Gemcitabine was chosen as the positive drug; the mice were then administered intravenous doses of **1c** (5 mg/kg), **1n** (5 mg/kg), **1a** (5 mg/kg), and gemcitabine (20 mg/kg) every other day for 24 days. Based on the monitoring of mouse body weight, **1n** had a lesser impact on weight gain compared to **1a**, indicating that **1n** may possess a higher safety profile (Figure 1C). Measurements of tumor volume and tumor weight revealed that **1n** demonstrated a tumor inhibition rate comparable to that of gemcitabine. In contrast, although the **1a**-treated group exhibited a high tumor inhibition rate, it was accompanied by a 50% mortality rate among the mice (Figure 1D−G).

In the biochemical and histological assays, **1n** (5 mg/kg) and **1a** (5 mg/kg) were administered to mice every other day, with the control group receiving an equivalent dose of saline. Following 14 days, mouse serum was collected for biochemical analysis, and organs were fixed for histological examination. Accordingly, serum biochemical analysis illustrated that ALT and AST levels were substantially elevated in the **1a**-treated group compared to the control group, indicating its potential hepatotoxicity. In contrast, no significant differences were observed between the **1N**-treated group and the control group (Figure 2A). Histological examination of liver and kidney tissues using H&E staining revealed that the liver tissues in the **1a**-treated group showed signs of hepatocyte necrosis and disruption of the hepatic cord structure. In comparison, the liver tissues of the **1n**-treated and control groups were normal. Additionally, H&E staining of kidney tissues revealed no significant pathological differences in any group (Figure 2B).

## 3. Materials and Methods

### 3.1. Chemistry

Reagents were purchased from the Experimental Reagents and Technology Management Platform of Nankai University without further purification, unless otherwise stated. Dry solvents such as tetrahydrofuran (THF), dimethylformamide (DMF), and dichloromethane (DCM) were purified by solvent purification systems (Innovative Technology) or purchased directly as ultradry solvents. Reactions were monitored by thin layer chromatography (TLC) performed on silica gel plates using UV light as a visualizing agent and aqueous phosphomolybdic acid or basic aqueous potassium permanganate as a developing agent; 200–300 mesh silica gel was used for column chromatography.

^1^H NMR (TMS as the internal standard), ^19^F NMR spectra (CFCl_3_ as the outside standard and low field is positive), and ^13^C NMR spectra were recorded on a Bruker AV 400 instrument and calibrated by using internal references and solvent signals CDCl_3_ (*δ*_H_ 7.26, *δ*_C_ 77.16), DMSO-*d*6 (*δ*_H_ 2.50, *δ*_C_ 39.52), or Acetone-*d*6 (*δ*_H_ 2.05, *δ*_C_ 206.26, 29.84). ^1^H NMR data are reported as follows: chemical shift, multiplicity (s, singlet; d, doublet; t, triplet; q, quartet; br, broad; m, multiplet), coupling constants, and integration. High-resolution mass spectra (HRMS) were obtained with a Thermo ScientificTM Q Exactive Focus mass spectrometer (Orbitrap spec). Optical rotations were recorded on an Insmark IP 120 digital polarimeter. The synthesis of compounds **1a**–**13a** and **2b** has been reported in previous studies [19].

Chemistry experimental procedures, except for compound **8n**, are available in the Appendix A, which include synthetic procedures and characterization data (pages 5–59), NMR spectra (Appendix A), and HPLC analysis (Appendix A).

#### 3.1.1. General Procedure for the Synthesis of **8b**–**8j** and **8n** (Take **8n** for Example)

To the solution of **6c** (3.23g, 6.58 mmol) in THF (20 mL) was added 4 N HCl in dioxane (32.9 mL, 131.70 mmol) at 0 °C. The mixture was allowed to reach room temperature, stirred for 1 h, and concentrated under reduced pressure to provide a white solid. To this salt was added a solution of **7d** (1.96 g, 6.58 mmol) and HATU (3.76 g, 9.88 mmol) in DCM (50 mL), then DIPEA (3.3 mL, 19.75 mmol) was added dropwise at 0 °C. The reaction was allowed to reach room temperature and stirred overnight. The mixture was concentrated, diluted with EtOAc (100 mL), and washed with aqueous 5% NaHSO_4_ (3 × 100 mL) and brine (100 mL), dried (Na_2_SO_4_), filtered, concentrated in vacuo, and purified by column chromatography (25/1 to 2/1 petroleum ether/EtOAc) to provide compound **8n** (3.31 g, 75%) as a white solid. $[\alpha {]}_{D}^{22}$ = −87.7 (*c* = 0.5, CHCl_3_). ^1^H NMR (400 MHz, CDCl_3_) *δ* 7.30 (d, *J* = 2.7 Hz, 5H), 7.13 (m, 2H), 7.00–6.65 (m, 3H), 5.20–5.02 (m, 2H), 5.01 (d, *J* = 7.9 Hz, 1H), 4.83 (dd, *J* = 29.8, 9.3 Hz, 2H), 4.50 (*p*, *J* = 6.4 Hz, 1H), 3.27 (dd, *J* = 14.4, 6.4 Hz, 1H), 3.02 (s, 1H), 2.97–2.82 (m, 3H), 2.71 (d, *J* = 25.0 Hz, 3H), 2.27–2.13 (m, 1H), 1.35 (d, *J* = 18.2 Hz, 9H), 1.18 (d, *J* = 6.3 Hz, 3H), 1.02–0.93 (m, 3H), 0.79–0.67 (m, 3H). ^13^C NMR (100 MHz, CDCl_3_) *δ* 170.2, 169.3, 163.0, 160.5, 156.2, 135.4, 133.2, 130.7, 130.6, 130.5, 128.7, 128.7, 128.6, 128.6, 128.5, 128.2, 122.9, 120.3, 115.3, 115.1, 80.8, 75.2, 67.1, 66.9, 61.9, 59.8, 52.3, 52.2, 33.3, 32.0, 31.6, 31.0, 29.8, 29.7, 29.4, 28.3, 28.2, 28.1, 28.1, 27.3, 22.8, 19.8, 19.7, 18.7, 18.6, 17.3, 14.2. ^19^F NMR (376 MHz, CDCl_3_) *δ*−57.9, −58.3, −58.4, −116.3, −116.6 (rotamer). HRMS (ESI) *m*/*z*: calculated for C_33_H_43_N_3_O_7_F_4_Na^+^ [M + Na]^+^: 692.2929, found: 692.2923.

#### 3.1.2. General Procedure for the Synthesis of **3b**–**3j** and **3n** (Take **3n** for Example)

To the solution of 8n (4.70 g, 7.02 mmol) in EtOH (50 mL) was added Pd/C 10% Wt (470 mg) at 0 °C. The reaction vessel was evacuated/backfilled with argon three times first, and then the reaction vessel was evacuated/backfilled with H_2_ three times and stirred under this atmosphere overnight. The mixture was filtered through a pad of celite, washed with MeOH, and concentrated in vacuo and purified by column chromatography (100/1 to 25/1 DCM/MeOH) to provide compound 3n (3.50 g, 86%) as a white solid. $[\alpha {]}_{D}^{22}$ = −87.3 (*c* = 0.5, CHCl_3_). ^1^H NMR (400 MHz, CDCl_3_) *δ* 7.60 (s, 1H), 7.18–7.07 (m, 3H), 6.98–6.83 (m, 2H), 5.18–5.05 (m, 1H), 4.90–4.72 (m, 2H), 4.57 (t, *J* = 6.5 Hz, 1H), 3.25 (dd, *J* = 14.4, 7.0 Hz, 1H), 3.06 (d, *J* = 46.7 Hz, 3H), 2.89 (d, *J* = 5.5 Hz, 1H), 2.74 (d, *J* = 21.3 Hz, 3H), 2.31–2.12 (m, 1H), 1.33 (d, *J* = 23.8 Hz, 12H), 1.09–0.98 (m, 3H), 0.76 (dd, *J* = 33.2, 6.1 Hz, 3H). ^13^C NMR (100 MHz, CDCl_3_) *δ* 173.6, 172.1, 170.7, 170.4, 169.6, 169.3, 163.0, 160.6, 156.6, 156.3, 133.0, 130.8, 130.7, 130.5, 125.4, 122.9, 120.4, 115.6, 115.4, 115.2, 81.3, 81.1, 75.2, 65.0, 62.4, 61.1, 59.7, 59.6, 52.4, 33.5, 33.1, 32.3, 31.0, 30.4, 29.8, 29.6, 28.2, 28.2, 27.5, 27.3, 19.9, 19.7, 18.8, 18.2, 17.5. ^19^F NMR (376 MHz, CDCl_3_) *δ* −57.9, −58.2, −116.3, −116.6. (rotamer). HRMS (ESI) *m*/*z*: calculated for C_26_H_37_N_3_O_7_F_4_Na^+^ [M + Na]^+^: 602.2460, found: 602.2448.

#### 3.1.3. General Procedure for the Synthesis of **13b**–**13n** (Take **13n** for Example)

To the solution of 4a (8.3 g, 19.53 mmol) in DCM (50 mL) was added 4 N HCl in dioxane (97.7 mL, 390.70 mmol) at 0 °C. The mixture was allowed to reach room temperature, stirred for 30 min, and concentrated under reduced pressure to provide a white solid. To this salt was added a solution of 3n (10.19 g, 17.58 mmol) and HATU (11.14 g, 19.30 mmol) in DCM (50 mL), then DIPEA (9.7 mL, 58.60 mmol) was added dropwise at 0 °C. The reaction was allowed to reach room temperature and stirred overnight. The mixture was concentrated, diluted with EtOAc (100 mL), and washed with aqueous 5% NaHSO_4_ (3 × 100 mL) and brine (100 mL), dried (Na_2_SO_4_), filtered, concentrated in vacuo, and purified by column chromatography (3/1 to 1/2 DCM/EtOAc) to provide compound 13n (7.42 g, 55%) as a white foamed solid. $[\alpha {]}_{D}^{22}$ = −105.3 (*c* = 0.5, CHCl_3_). ^1^H NMR (400 MHz, CDCl_3_) *δ* 7.28 (d, *J* = 2.0 Hz, 1H), 7.17–7.12 (m, 2H), 7.02–6.80 (m, 3H), 6.08 (dd, *J* = 6.0, 1.6 Hz, 1H), 5.59 (dd, *J* = 10.1, 2.0 Hz, 1H), 5.10–4.99 (m, 2H), 4.85–4.71 (m, 2H), 4.57 (q, *J* = 6.0 Hz, 1H), 4.37 (s, 1H), 3.92 (d, *J* = 11.6 Hz, 1H), 3.82 (dd, *J* = 11.6, 4.4 Hz, 1H), 3.29 (dd, *J* = 14.4, 6.4 Hz, 1H), 3.12–2.90 (m, 4H), 2.75 (d, *J* = 39.8 Hz, 4H), 2.39 (ddd, *J* = 14.6, 10.1, 4.7 Hz, 1H), 2.24 (s, 1H), 2.02 (d, *J* = 13.4 Hz, 1H), 1.45 (d, *J* = 6.7 Hz, 3H), 1.37 (d, *J* = 20.7 Hz, 9H), 1.32 (d, *J* = 2.7 Hz, 3H), 0.99 (d, *J* = 6.3 Hz, 3H), 0.85–0.69 (m, 3H). ^13^C NMR (100 MHz, CDCl_3_) *δ* 174.6, 170.4, 170.0, 169.0, 163.0, 160.6, 156.2, 154.4, 133.3, 130.7, 130.7, 125.4, 122.9, 120.3, 115.4, 115.2, 80.9, 74.9, 71.9, 60.3, 59.5, 58.6, 58.3, 57.2, 52.5, 52.2, 38.8, 36.7, 33.4, 31.6, 30.6, 29.8, 28.3, 27.4, 18.9, 18.4, 17.9, 17.0. ^19^F NMR (376 MHz, CDCl_3_) *δ*−58.3, −58.5, −116.3, −116.6 (rotamer). HRMS (ESI) *m*/*z*: calculated for C_36_H_49_N_5_O_9_F_4_Na^+^ [M + Na]^+^: 794.3359, found: 794.3352.

#### 3.1.4. General Procedure for the Synthesis of **1b**–**1n** (Take **1n** for Example)

To the solution of 13n (7.42 g, 9.61 mmol) in DCM (50 mL) was added 4 N HCl in dioxane (96.1 mL, 384.60 mmol) at 0 °C. The mixture was allowed to reach room temperature, stirred for 2 h, and concentrated under reduced pressure to provide a white solid. To this salt was added a solution of 2b (4.56 g, 28.84 mmol) and HATU (10.97 g, 28.84 mmol) in DCM (50 mL), then DIPEA (9.5 mL, 57.68 mmol) was added dropwise at 0 °C. The reaction was allowed to reach room temperature and stirred overnight. The mixture was concentrated, diluted with EtOAc (30 mL), and washed with aqueous 5% NaHSO_4_ (3 × 100 mL) and brine (100 mL), dried (Na_2_SO_4_), filtered, concentrated in vacuo, and purified by column chromatography (2/1 to 1/3 DCM/EtOAc) to provide compound 1n (4.83 g, 62%) as a white foamed solid. $[\alpha {]}_{D}^{22}$ = −122.7 (*c* = 0.5, CHCl_3_). ^1^H NMR (400 MHz, CDCl_3_) *δ* 7.28 (d, *J* = 2.0 Hz, 1H), 7.20–7.10 (m, 2H), 6.97–6.87 (m, 3H), 6.07 (dd, *J* = 6.1, 1.6 Hz, 1H), 5.58 (dd, *J* = 10.1, 2.1 Hz, 1H), 5.46 (dd, *J* = 9.8, 6.6 Hz, 1H), 5.09–4.98 (m, 2H), 4.79 (qt, *J* = 6.7, 1.8 Hz, 1H), 4.53 (*p*, *J* = 6.2 Hz, 1H), 4.36 (d, *J* = 4.7 Hz, 1H), 3.91 (d, *J* = 11.9 Hz, 1H), 3.84 (dd, *J* = 11.6, 4.4 Hz, 1H), 3.22 (dd, *J* = 14.8, 6.5 Hz, 1H), 3.00 (s, 3H), 2.95 (d, *J* = 5.1 Hz, 1H), 2.88 (s, 3H), 2.53 (h, *J* = 6.8 Hz, 1H), 2.39 (ddd, *J* = 14.6, 10.1, 4.8 Hz, 1H), 2.22 (dq, *J* = 10.9, 6.6 Hz, 1H), 2.01 (dd, *J* = 14.5, 1.9 Hz, 1H), 1.45 (d, *J* = 6.8 Hz, 3H), 1.42–1.36 (m, 1H), 1.30 (d, *J* = 6.2 Hz, 3H), 1.25–1.09 (m, 8H), 1.04 (d, *J* = 6.7 Hz, 3H), 0.99 (d, *J* = 6.5 Hz, 3H), 0.97–0.90 (m, 2H), 0.86 (t, *J* = 7.2 Hz, 3H), 0.75 (d, *J* = 6.7 Hz, 3H). ^13^C NMR (100 MHz, CDCl_3_) *δ* 178.2, 174.4, 170.1, 169.9, 169.0, 168.9, 168.8, 168.8, 162.9, 160.5, 154.3, 132.5, 132.5, 130.5, 130.4, 125.3, 122.7, 120.2, 115.4, 115.2, 74.8, 74.8, 71.7, 59.3, 58.5, 58.2, 57.0, 56.9, 52.2, 36.6, 36.1, 34.0, 32.8, 31.6, 30.9, 30.5, 29.2, 27.3, 27.1, 22.6, 18.8, 18.4, 17.9, 16.9, 14.0. ^19^F NMR (376 MHz, CDCl_3_) *δ* −58.3, −116.2. HRMS (ESI) *m*/*z*: calculated for C_40_H_57_N_5_O_8_F_4_Na^+^ [M + Na]^+^: 834.4035, found: 834.4029.

### 3.2. Experimental Cells and Animals

PANC-1 was purchased from the National Experimental Cell Resource Sharing Platform. PANC-1 was cultured in DMEM medium containing 10% serum. The 4-week-old BALB/C nude mice were purchased from Beijing Vital River Laboratory Animal Technology Co., Ltd. and raised at the Animal Resources Center of Nankai University. The animal experiments have been approved by the Ethics Committee of Nankai University (2022-SYDWLL-000025, 2024-SYDWLL-000762).

### 3.3. MTT Cell Activity Assay

A total of 3000 PANC-1 cells were resuspended in 200 μL of culture medium and seeded into 96-well plates. After 24 h, the cells were treated with the compound and incubated for an additional 72 h. Following the incubation, 20 μL of 5 mg/mL MTT solution (Solarbio, China) was added to each well in the dark, and the cells were incubated for another 4 h. After incubation, the culture medium was discarded, and the wells were washed with PBS to remove any residual medium. Then, 200 μL of DMSO was added to each well to fully dissolve the formazan crystals, and the solution was mixed thoroughly. The absorbance was measured at 492 nm/570 nm, and the data were analyzed using GraphPad Prism.

### 3.4. Stability in Mouse Plasma Assay

Thaw plasma at room temperature and vortex for 3–5 s. Centrifuge at 3000 rpm at 10 °C for 10 min, and carefully transfer the supernatant to a new Eppendorf tube. Preincubate the plasma at 37 °C for 10 min. Prepare a series of test substances by adding 3 μL of the compound (100 μM) to 297 μL of plasma to achieve a final concentration of 1 μM, ensuring that the DMSO concentration in plasma is 1%. Propantheline bromide is used as a positive control. Incubate the mixture at 37 °C, and at specified time points (0, 1, 2, 4, and 24 h), take 50 μL of the plasma sample. To terminate the reaction, add 250 μL of acetonitrile (containing an internal standard), vortex thoroughly, and centrifuge at 16,000 rcf (4 °C) for 10 min. Collect 130 μL of the supernatant and dilute with 70 μL of pure water for LC-MS/MS analysis. The concentration of the test substances in plasma samples was determined using LC-MS/MS. The stability of the test substance in plasma was evaluated by measuring the remaining percentage of the compound in the plasma over time. For example, the remaining percentage of bromopropyl in the plasma of CD-1 mice after a 2-h incubation was 0.90%, demonstrating that the testing system is suitable for evaluating the stability of test substances in plasma.

### 3.5. In Vivo Antitumor Activity

A total of 4 × 10^6^ PANC-1 cells suspended in 100 μL PBS were subcutaneously injected into the right shoulder region of BALB/c nude mice. When the tumor volume reached approximately 200 mm^3^, the mice were euthanized, and the tumor tissues were harvested and designated as F1 generation. Tumor tissues were then cut into small cubes measuring approximately 3 × 3 × 3 mm, which were implanted into new BALB/c mice to propagate the tumors. This procedure was repeated to obtain the F2 generation of tumors. Finally, mice bearing F3 generation transplant tumors were used for drug treatment experiments. When the tumor volume reached approximately 80–100 mm^3^, the mice were randomly divided into five groups (*n* = 6): control group, 5 mg/kg compound **1a** group, 5 mg/kg compound **1c** group, 5 mg/kg compound **1n** group, and 20 mg/kg gemcitabine group. The compounds **1a**, **1c**, and **1n** were dissolved in DMSO and then diluted with 0.9% physiological saline to achieve a final concentration of 0.5 mg/mL as the dosing concentration. Gemcitabine was dissolved directly in 0.9% saline to achieve a final concentration of 2 mg/mL as the dosing concentration. The compounds were administered intravenously every two days, alternating between the left and right tail veins to minimize vein damage. Injections were performed using ultra-fine syringes (30G needles) to reduce stress and ensure consistent delivery. The body weight and tumor volume of the mice were measured every four days. On day 24, the mice were euthanized, and the tumor weights were recorded. Tumor volume (V) was calculated using the formula: V = (ab^2^)/2, with “a” representing the long diameter of the tumor and “b” representing the short diameter of the tumor. Tumor volume measurements showed that the **1n**-treated group exhibited a tumor inhibition rate comparable to that of gemcitabine, with tumor volumes of 553.3 ± 166.8 mm^3^ and 561.1 ± 177.3 mm^3^, respectively. The **1a**-treated group demonstrated a significantly smaller tumor volume of 353.0 ± 66.7 mm^3^, indicating a high tumor inhibition rate, although this group experienced a 50% mortality rate among mice. In contrast, the control group displayed a much larger tumor volume of 1262 ± 137.7 mm^3^ (Figure 1D). Similarly, tumor weight measurements supported these findings. The **1n** and gemcitabine-treated groups showed comparable tumor weights of 0.542 ± 0.156 g and 0.548 ± 0.153 g, respectively. The **1a**-treated group had a significantly lower tumor weight of 0.393 ± 0.049 g, while the control group exhibited the highest tumor weight of 1.118 ± 0.177 g (Figure 1E,F). Tumor weight was used to calculate the tumor inhibition rate (Figure 1G).

### 3.6. H&E Staining

Mice were euthanized, and liver and kidney tissues were collected and fixed in 10% neutral-buffered formalin for 48 h. The fixed tissues were processed for paraffin embedding, including dehydration in a graded ethanol series, clearing with xylene, and infiltration with molten paraffin wax. Paraffin blocks were sectioned into 4 μm slices using a rotary microtome, and the sections were mounted onto glass slides. The slides were dried at 60 °C for 4 h to ensure proper tissue adhesion. Following deparaffinization and rehydration, staining was performed according to the manufacturer’s instructions for the H&E staining kit (Solarbio, China). The sections were immersed in a hematoxylin solution for 3 min to stain the nuclei and then rinsed under running tap water for 1 minute to remove excess stain. Differentiation was achieved by briefly immersing the slides in a 1% hydrochloric acid-ethanol solution, followed by neutralization in 0.6% ammonia water until the nuclei appeared blue. Eosin staining was applied for 30 s to stain cytoplasmic and extracellular components, imparting a pink coloration. The slides were rinsed under running tap water for 30 s to remove excess eosin. Finally, the sections were dehydrated through a graded ethanol series, cleared with xylene, and coverslipped with a neutral resin mounting medium to preserve the stained tissues for microscopic examination.

## 4. Conclusions

The syntheses of 26 majusculamide D analogues were successfully completed and subsequently evaluated for their anti-pancreatic cancer activity. Among them, two compounds (**1i** and **1j**) demonstrated IC_50_ values < 1 nM. The biological results indicate that the C23-OAc could be replaced with OCF_3_. The removal of the C4-Me group and the acetyl group of C23-OAc both resulted in a significant decrease in activity. However, replacing the C10-OH and C34-OMe with an F atom was well tolerated. Although attaching one extra amino acid at C23 enhanced the water solubility of natural products, their activities diminished dramatically. A series of in vitro assays, including wound healing, transwell, clone formation, EdU, and western blot, were conducted for the purpose of evaluating the inhibitory effects of majusculamide D on migration, invasion, and proliferation against pancreatic cancer cell lines. The results indicated that majusculamide D inhibited the migration and proliferation of pancreatic cancer cells and may be involved in the epithelial-mesenchymal transition (EMT) process.

The designed analog **1n** exhibited an apparent improvement in the mouse plasma stability assay compared to majusculamide D. The in vivo data of **1n** demonstrated therapeutic benefit compared to the control group. Above all, **1n** overcame the severe toxicity of mouse death arising in the natural product-treated group at a dosage of 5 mg/kg. The serum biochemical analysis indicated potential hepatotoxicity of the natural product, as evidenced by elevated ALT and AST levels. Histological examination of liver and kidney tissues with H&E staining further confirmed hepatocyte necrosis and disruption of hepatic cord structure in the natural product-treated group. In contrast, the **1n**-treated group showed no significant differences in serum biochemical parameters compared to the control group, and liver tissue appeared normal. The results collectively suggested that natural products may have higher toxicity, and our modification effectively reduced the severe toxicity, such as mouse death, of natural products while maintaining a good tumor inhibition rate. However, the current results only demonstrate that **1n** is less toxic than natural products, with the underlying mechanisms remaining unclear. Further studies, particularly on its exact mode of action and pharmacokinetics, are necessary. Efforts to expand the therapeutic window through strategies such as prodrugs, antibody-drug conjugates, peptide-drug conjugates, and other innovative approaches are ongoing, and additional findings will be reported in due course.

## Data Availability

The data are contained within the article or Appendix A; further inquiries can be directed to the corresponding author.

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
