# Peer review of "Structure-Activity Relationship Study of Majusculamide D: Overcoming Metabolic Instability and Severe Toxicity with a Fluoro Analogue"

_marinedrugs, 2024, doi:10.3390/md22120537_

Round 1

Reviewer 1 Report

Comments and Suggestions for Authors

Dear Authors, please see comments below:

The article describes that Pancreatic cancer patients have poor prognosis, with 80-85% unresectable or metastasis. Drugs to halt metastasis are needed. The study synthesized 17 majusculamide D analogues, two of which showed IC50 values below 1 nM against PANC-1 cancer cells. In vitro assays confirmed inhibition of cell migration, invasion, and proliferation. The optimized fluorinated analogue improved plasma stability and tumor suppression. The article is interesting and will contribute anticancer studies using novel marine compounds.

1.        The abstract is concise and covers all aspects of the research suitably.

2.        The introduction has a good background of the literature. However, additional information on Lyngbya majuscule is encouraged. Please expand on the anticancer activities against various types of cancer cells in vitro

3.        Results is very well presented; however, the images and graphs must be improved because it is not clear.

4.        Please indicate that the results and discussion have been combined by including “discussion “ next to the subheadings of results.

5.        Materials and methods are very descriptive.

6.        Please explain briefly the future studies

Author Response

Comments and Suggestions for Authors

Dear Authors, please see comments below:

The article describes that Pancreatic cancer patients have poor prognosis, with 80-85% unresectable or metastasis. Drugs to halt metastasis are needed. The study synthesized 17 majusculamide D analogues, two of which showed IC50 values below 1 nM against PANC-1 cancer cells. In vitro assays confirmed inhibition of cell migration, invasion, and proliferation. The optimized fluorinated analogue improved plasma stability and tumor suppression. The article is interesting and will contribute anticancer studies using novel marine compounds.

We thank the reviewer for the relevant and useful comments. Following is our replies to the reviewer’s comments.

1. The abstract is concise and covers all aspects of the research suitably.

Response: We thank you for the helpful comments.

2. The introduction has a good background of the literature. However, additional information on Lyngbya majuscule is encouraged. Please expand on the anticancer activities against various types of cancer cells in vitro.

Response: Thanks. In fact, we have completed MTT testing of Majusculamide D on different cancer cell lines in previous work, and the results are shown in Table S2 of the supporting information (https://doi.org/10.1002/cjoc.202300526), as shown in the following table. We have added an sentence “Besides, majusculamide D was found to show inhibitory activities against several cancer cell lines [19].” in the introduction part.

Table R1. Activities of Majusculamide D (1a) against different cancer cell lines and normal cell line

IC50(nM)

A549

U251

H820

HeLa

U87

SMMC-7721

HCT116

Beas-2B

Majusculamide D

2.5±0.5

2.4±0.3

78.9±7.6

1.9±0.4

4.4±1.0

2.3±1.4

3.0±1.2

12.0±3.9

3. Results is very well presented; however, the images and graphs must be improved because it is not clear.

Response: Thanks. We have adjusted the images and graphs to 300 dpi.

4. Please indicate that the results and discussion have been combined by including “discussion” next to the subheadings of results.

Response: Thanks. We have added the “discussion” next to the subheadings of results.

5. Materials and methods are very descriptive.

Response: We thank you for the helpful comments.

6. Please explain briefly the future studies

Response: Thanks. In the conclusions section, we have added sentence “...1n overcame the severe toxicity like mouse death arising in the natural product-treated group at dosage of 5 mg/kg and is worth further exploration, including its exact mode of action, pharmacokinetics, etc. The therapeutic window further broadened by employing strategies such as prodrugs, antibody-drug conjugates, peptide-drug conjugates, and other innovative approaches are ongoing. Further results will be reported in due course.” 

Reviewer 2 Report

Comments and Suggestions for Authors

This is a novel drug improvement research. However, the word toxicity in the title is not justified.

The controls are missing throughout the manuscript. There was no other cell line used as a control. Methods state 3K cells but do not mention which cells

The gel blots provided have no figure with labelling, and it's difficult to follow the inferences drawn. 

The supplementary file is too stand alone and lacks cross citation in the main manuscript. 

The compound dosage calculation is not explained nor the serum levels are monitored. Liver and kidney histopathology or any enzyme based toxicity studies were not performed. 

There is no data as to how these derivatives were less toxic. Were normal tissue organoid controls used vs. cancerous? There is lack of information and the manuscript lacks all over cohesiveness

Comments on the Quality of English Language

The language needs editing. Sentences beginning with numbers aren't a norm for publishing. The paragraph layout also lacks a flow

Author Response

Comments and Suggestions for Authors

Point 1: This is a novel drug improvement research. However, the word toxicity in the title is not justified.

Response: Thank you for the helpful comments. In the in vivo antitumor experiment, all six mice survived in the compound 1n-treated group while only half of the mice survived in the majusculamide D-treated (1a) group. Besides, we conducted serum biochemical analysis. The results indicated that ALT and AST levels were substantially elevated in the 1a-treated group compared to the control group, suggesting potential hepatotoxicity of 1a. In contrast, no significant differences were observed between the 1n-treated group and the control group (Figure R1). Further histological examination of liver and kidney tissues using H&E staining revealed that the liver tissues in the 1a-treated group showed signs of hepatocyte necrosis and disruption of the hepatic cord structure. In comparison, the liver tissues of the 1n-treated and control groups were normal (Figure R2). The results collectively suggested that 1a may have higher toxicity, and our modification effectively reduced the severe toxicity such as mouse death of 1a while maintaining a good tumor inhibition rate. 

Figure R1. Serum biochemical analysis of mice administered with 1a and 1n.

Figure R2. Representative images of hematoxylin and eosin (H&E) staining of liver and kidney tissues from mice administered with 1a and 1n.

Point 2: The controls are missing throughout the manuscript. There was no other cell line used as a control. Methods state 3K cells but do not mention which cells

Response: Thank you for the helpful comment. In the MTT tests (Table 1 and Table 2), the parent compound majusculamide D was selected as the control. We have conducted MTT tests of majusculamide D against different cancer cell lines in previous work, and the results are shown in Table S2 of the supporting information (https://doi.org/10.1002/cjoc.202300526). The results are also shown in the following Table R1. Meanwhile, we provided a detailed description of the cell type as "PANC-1" in MTT cell activity assay

Table R1. Activities of Majusculamide D (1a) against different cancer cell lines and normal cell line

IC50(nM)

A549

U251

H820

HeLa

U87

SMMC-7721

HCT116

Beas-2B

Majusculamide D

2.5±0.5

2.4±0.3

78.9±7.6

1.9±0.4

4.4±1.0

2.3±1.4

3.0±1.2

12.0±3.9

Point 3: The gel blots provided have no figure with labelling, and it's difficult to follow the inferences drawn.

Response: Thank you for the helpful comment. We deeply apologize for any inconvenience. The  figure of western blot with labelling are shown in Supplementary Materials Figure S225 (E). We previously uploaded a separate file without biomakers because the submission required us to upload the original, uncropped and unadjusted images supporting all blot and gel results reported. 

Figure R3. Western blot assays of 1a on biomarkers

Point 4: The supplementary file is too stand alone and lacks cross citation in the main manuscript.

Response: Thank you for the helpful comment. We have made the corresponding revisions. 

Point 5: The compound dosage calculation is not explained nor the serum levels are monitored. Liver and kidney histopathology or any enzyme based toxicity studies were not performed.

Response: Thank you for the helpful comment. The dosage of 5 mg/kg is an effective dosage we summarized after two-time mice experiments. According to your suggestion, we further conducted biochemical and histological assays using mice. Compounds 1n (5 mg/kg) and 1a (5 mg/kg) were administered every other day, with the control group receiving an equivalent dose of saline. Following 14 days, mouse serum was collected for biochemical analysis, and organs were fixed for histological examination. Accordingly, serum biochemical analysis illustrated that ALT and AST levels were substantially elevated in the 1a-treated group compared to the control group, indicating its potential hepatotoxicity. In contrast, no significant differences were observed between the 1n-treated group and the control group (Figure R1). Histological examination of liver and kidney tissues using H&E staining revealed that the liver tissues in the 1a-treated group showed signs of hepatocyte necrosis and disruption of the hepatic cord structure. In comparison, the liver tissues of the 1n-treated and control groups were normal. Additionally, H&E staining of kidney tissues revealed no significant pathological differences in any group (Figure R2).

Figure R1. Serum biochemical analysis of mice administered with 1a and 1n.

Figure R2. Representative images of hematoxylin and eosin (H&E) staining of liver and kidney tissues from mice administered with 1a and 1n.

Point 6: There is no data as to how these derivatives were less toxic. Were normal tissue organoid controls used vs. cancerous? There is lack of information and the manuscript lacks all over cohesiveness

Response: Thank you for the helpful comment. Unfortunately, we don’t have the experimental conditions to establish organoid models. In the in vivo antitumor experiment, all six mice survived in the compound 1n-treated group while only half of the mice survived in the majusculamide D-treated (1a) group. Besides, we conducted serum biochemical analysis. The results indicated that ALT and AST levels were substantially elevated in the 1a-treated group compared to the control group, suggesting potential hepatotoxicity of 1a. In contrast, no significant differences were observed between the 1n-treated group and the control group (Figure R1). Further histological examination of liver and kidney tissues using H&E staining revealed that the liver tissues in the 1a-treated group showed signs of hepatocyte necrosis and disruption of the hepatic cord structure. In comparison, the liver tissues of the 1n-treated and control groups were normal (Figure R2). The results collectively suggested that 1a may have higher toxicity, and our modification effectively reduced the severe toxicity such as mouse death of 1a while maintaining a good tumor inhibition rate. Actually, the experimental results only indicate that 1n is less toxic than 1a, its reason are still unclear and needs further elaboration, especially for pharmacokinetics study. These will be studied in the future. 

Comments on the Quality of English Language

The language needs editing. Sentences beginning with numbers aren't a norm for publishing. The paragraph layout also lacks a flow

Response: Thank you for the helpful comment. We have reorganized the layout of the paragraph, scheme and figure. This version is not the final version for publishing, it is just for reviewing, so the sentences begin with numbers. 

Reviewer 3 Report

Comments and Suggestions for Authors

This study on the design and pharmacological evaluation of analogues of majusculamide D is interesting but the manuscript is poorly written. The text is full of errors of language, making it hard to read. The language must be revised by a native-speaking English professional.

Comments:

1.       A poor Abstract. The first part of the abstract must be reconsidered. The general comments about pancreatic cancer should be removed (useless in the abstract). In contrast, it is important to define what is majusculamide D (origin, structural motif) and explain the overall design strategy. The abstract should provide also quantitative information for the best products.

2.       The chemical rational is not explained. 26 analogues were designed. The rational must be explained in the introduction. Why these 26 compounds? How they were selected? The chemical design/strategy is not clearly explained.

3.       Proper legends are missing for Schemes 2-3 and the Tables. Each Scheme/Figure must have a full legend to explain the synthetic scheme and to define precisely the measured parameters. (IC50 at which time? - solubility test?).

4.       Drug stability. Compound 1a is the most active compound in vivo, but the less stable metabolically. Compound 1n is more stable but significantly less active. These data could suggest that 1a is acting as a prodrug, transformed into an active product upon metabolization. Has this hypothesis been considered? And discussed.

5.       The chemistry data for the newly described compounds should be given in the main text in a concise form (not only as Supplementary Materials). The analytical data for the compounds tested in vivo should be included in the core of the manuscript.

6.       The recent literature on majusculamide should be integrated into the manuscript. The Discussion should evoke other recent studies devoted to the synthesis and identification of majusculamides, to include the study into a broader context. Mechanistic information are missing (potential drug targets, orientation/perspectives).

The manuscript is potentially acceptable for publication in the journal, pending a profound and careful revision.

Comments on the Quality of English Language

The English language must be improved

Author Response

Comments and Suggestions for Authors

This study on the design and pharmacological evaluation of analogues of majusculamide D is interesting but the manuscript is poorly written. The text is full of errors of language, making it hard to read. The language must be revised by a native-speaking English professional.

Response: We thank you for the helpful comments. The language have been polished by a English professional.

Comments:

1. A poor Abstract. The first part of the abstract must be reconsidered. The general comments about pancreatic cancer should be removed (useless in the abstract). In contrast, it is important to define what is majusculamide D (origin, structural motif) and explain the overall design strategy. The abstract should provide also quantitative information for the best products.

Response: We thank you for the helpful comment. We have made the corresponding revisions. Now the Abstract: Majusculamide D, isolated from the marine cyanobacterium Moorea producens, is an anticancer lipopentapeptede consisting of fatty acid, tripeptide and pyrrolyl proline moieties. In this work, by utilizing convergent synthetic approach, late-stage modification and bioisostere strategy, 26 majusculamide D analogues were synthesized, and two (1i and 1j) demonstrated IC50 values < 1 nM against PANC-1 cancer cells. The results summarized a preliminary structure-activity relationship mainly at the C23, C4, C34 and C10 sites. A series of in vitro assays, including wound healing, transwell, clone formation, EdU, and western blot, confirmed that majusculamide D inhibited the migration, invasion, and proliferation of pancreatic cancer cells. The optimized fluorinated analogue 1n demonstrated a notable enhancement in stability during the mouse plasma assay (> 50% left after 24 h), exhibited tumor-suppressive effects (51.5% at a dosage of 5 mg/kg), and successfully mitigated the severe toxicity (no mouse dead) observed in the group treated with majusculamide D (3 mice dead) in a xenografted mouse model.

2. The chemical rational is not explained. 26 analogues were designed. The rational must be explained in the introduction. Why these 26 compounds? How they were selected? The chemical design/strategy is not clearly explained.

Response: Thanks. We have made the corresponding revisions. We have revised and moved the sentences “To enhance the metabolic stability, it was anticipated that the C23-OAc (R1) of the tripeptide moiety would be replaced using the bioisostere strategy (e.g., NAc) [20]. The fluorine atom was considered to be introduced as it makes an impact on the potency, conformation, metabolism, and membrane permeability [21]. The para-substituted OMe (R3) group in the tyrosine of tripeptide was envisioned to be changed to F, H, OH, or Ph to evaluate its impact on the bioactivies. Regarding the pyrrolyl proline moiety, the configuration of C4 was intended to be switched, or the methyl group at C4 was to be directly trimmed, to investigate whether the naturally occurring C4-R-Me is necessary.  Given the poor water solubility of majusculamide D, we also aimed to improve its water solubility. The comprehensive SAR could guide us in further structural simplification and optimization [22]” to introduction part.

3. Proper legends are missing for Schemes 2-3 and the Tables. Each Scheme/Figure must have a full legend to explain the synthetic scheme and to define precisely the measured parameters. (IC50 at which time? - solubility test?).

Response: We thank you for the helpful comment. We have made the revison according to your suggestion. We have added the solubility test method, and the details of the IC50 testing are described in Section 3.3 titled "MTT Cell Activity Assay," which contains detailed instructions.

4. Drug stability. Compound 1a is the most active compound in vivo, but the less stable metabolically. Compound 1n is more stable but significantly less active. These data could suggest that 1a is acting as a prodrug, transformed into an active product upon metabolization. Has this hypothesis been considered? And discussed.

Response: We thank you for the helpful comment. The C23-OAc of 1a are supposed to bemetabolically instable since ester group are easily cleaved by esterase. We predicted that the possible metabolite of compound 1a would be 14a. However, 14a showed an significantly decreased inhibitory activity compared to 1a. We designed 1n with a C23-OCF3 group which was proved to be more metabolically stable than 1a.

5. The chemistry data for the newly described compounds should be given in the main text in a concise form (not only as Supplementary Materials). The analytical data for the compounds tested in vivo should be included in the core of the manuscript.

Response: We thank you for the helpful comment. We have included the synthetic details of compound 1n in the main text. The analytical illustration have been included in the Materials and Methods section.

6. The recent literature on majusculamide should be integrated into the manuscript. The Discussion should evoke other recent studies devoted to the synthesis and identification of majusculamides, to include the study into a broader context. Mechanistic information are missing (potential drug targets, orientation/perspe.

Response: Thanks for the suggestive comments. Research on majusculamide D is still relatively limited, especially regarding its mechanism of action, which will be the focus of our future efforts. Sentences “Above all, 1n overcame the severe toxicity like mouse death arising in the natural product-treated group at dosage of 5 mg/kg and is worth further exploration, including its exact mode of action, pharmacokinetics, etc. The therapeutic window further broadened by employing strategies such as prodrugs, antibody-drug conjugates, peptide-drug conjugates, and other innovative approaches are ongoing. Further results will be reported in due course.” have been included in the conclusions part.

The manuscript is potentially acceptable for publication in the journal, pending a profound and careful revision.

Response: Thanks for the comments.

Round 2

Reviewer 2 Report

Comments and Suggestions for Authors

Authors have given satisfactory responses to the comments raised in Round 1.

Authors need to clearly state the limitations described in the rebuttal in the discussion/conclusion sections of the manuscript.

Author Response

Authors have given satisfactory responses to the comments raised in Round 1.

Response: Thank you very much for your kind feedback and positive evaluation of our manuscript. We greatly appreciate your valuable comments and suggestions, which have helped us improve the quality of our work.

Authors need to clearly state the limitations described in the rebuttal in the discussion/conclusion sections of the manuscript.

Response: Thank you for the helpful comment. We have made the corresponding revisions in conclusion sections. 

Reviewer 3 Report

Comments and Suggestions for Authors

Some efforts have been made to improve the language and presentation of the manuscript. The revised manuscript can be accepted for publication in this journal.

Author Response

Some efforts have been made to improve the language and presentation of the manuscript. The revised manuscript can be accepted for publication in this journal.

Response: Thank you very much for your kind feedback and positive evaluation of our manuscript.